# Unraveling the phase diagram-ion transport relationship in aqueous electrolyte solutions and correlating conductivity with concentration and temperature by semi-empirical modeling

Hilal Al-Salih[1,2,3], Elena A. Baranova[1,2] & Yaser Abu-Lebdeh [3✉]

The relationship between structure and ion transport in liquid electrolyte solutions is not well understood over the whole concentration and temperature ranges. In this work, we have studied the ionic conductivity ($\kappa$) as a function of molar fraction (x) and Temperature (T) for aqueous solutions of salts with nitrate anion and different cations (proton, lithium, calcium, and ammonium) along with their liquid-solid phase diagrams. The connection between the known features in the phase diagrams and the ionic conductivity isotherms is established with an insight on the conductivity mechanism. Also, known isothermal ($\kappa$ vs.. x) and iso-compositional ($\kappa$ vs.. T) equations along with a proposed two variables semi-empirical model ($\kappa = f(x, T)$) were fitted to the collected data to validate their accuracy. The role of activation energy and free volume in controlling ionic conductivity is discussed. This work brings us closer to the development of a phenomenological model to describe the structure and transport in liquid electrolyte solutions.

[1] Department of Chemical and Biological Engineering, Centre for Catalysis Research and Innovation, University of Ottawa, 161 Louis-Pasteur, Ottawa, ON K1N 6N5, Canada. [2] Nexus for Quantum Technologies (NexQT), University of Ottawa, Ottawa, ON K1N 6N5, Canada. [3] Energy, Mining, and Environment Research Centre, National Research Council of Canada, 1200 Montreal Road, Ottawa, ON K1A 0R6, Canada. ✉email: Yaser.Abu-Lebdeh@nrc-cnrc.gc.ca

The exact structure of liquid electrolyte solutions is still not known due to their very disordered nature despite recent advancements in characterization techniques[1]. In their review of the structure of electrolyte solutions, Enderby and Neilson[2] highlighted the challenges in understanding the microstructure of liquid solutions, a sentiment shared by many scientists at the time. C. Angel, one prominent scientist in the field, described the lack of a model or theory that can successfully extend or replace the long-standing Debye-Hückel theory, which only applies to very low concentrations (mM), as the most celebrated failure in physical chemistry[3]. However, over the years there have been many serious attempts to extend and apply the theory to higher concentrations but were not overly successful. This topic has become of great interest recently as high concentrations (c) or molar fractions (x) of electrolyte solutions are required so that practical high ionic conductivities of $10^{-3}$–$10^{-2}$ S cm$^{-1}$ are achieved which often occurs at a concentration or molar fraction of highest (maximum) conductivity ($c_{max}$ or $x_{max}$) in the isothermal conductivity vs. composition plots. This concentration has not been given much significance in the past but recently a lot of research is focused on electrolyte solutions with higher concentrations beyond $x_{max}$ due to improved physiochemical and electrochemical performance[4,5].

It was shown by many researchers that the structure of liquid electrolyte solutions depends on the type of the salt and solvent and their concentration and temperature[6]. It is widely accepted that at very low concentrations ions are well separated into free ions while at low concentrations the salt dissociates into ions that are separated into solvated and unsolvated ion pairs (IP) of different types (contact, solvent separated and solvent shared) in a bulk solvent. As the concentration is increased fewer solvent molecules are available so the IPs can associate into larger ion clusters (ICs). Figure 1 shows possible structures of the different IPs and ICs that can exist in electrolyte solutions at different concentrations. This change in ion speciation has an impact on transport properties of the solution. Ionic conductivity in particular is one of the most important transport properties of liquid electrolyte solutions that qualify them for applications in many electrochemical devices.

Abu-Lebdeh et al[7] have recently showed that the strong dependence of ionic conductivity on electrolyte concentration in aqueous and non-aqueous solutions can be correlated to the liquid -solid (salt/solvent) binary phase diagram. It was shown that the presence of maxima in the bell-shaped, isothermal conductivity vs.. concentration plots is a common feature in liquid electrolyte solutions and also solid electrolytes[5]. A strong correlation between the molar fraction of the first eutectic point in phase diagrams and the molar fraction of the maximum conductivity on the conductivity isotherms was observed. Angell[3] proposed this to be a simple balance between concentration and mobility of ions while Abu-Lebdeh[8], gave this phenomena higher significance and attributed it to a transition in the structure of the solution from the ionic atmosphere structure where transport is dominated by solvated ionic species (different types of IPs) diffusing or migrating by a vehicular mechanism through free volume to a loose lattice structure where transport is dominated by ion hopping mechanism where naked ions hop along the extended structure (different types of ICs) from one site to another energetically favorable vacant site (free volume) (Fig. 1). Both mechanisms often depend on temperature, as higher temperatures can provide the energy needed for ions to overcome energy barriers and move through the free volume that in turn can expand with temperature. The energy barriers are directly linked to the charge density of the ions and their ability to attract and retain solvent molecules that can be quantified by the Eigen number in the case of water as a solvent[9,10].

Abu-Lebdeh proposed a model for electrolyte solutions where the change in liquid structure and hence transport mechanism can be understood by observing changes in solid microstructure in the (salt/solvent) phase diagram for a given concentration or molar fraction[7,11,12]. It assumes that, at the sub-micron level, the structure of the liquid, above the liquidus line, is a heterogeneous mixture of molten domains similar to that of the solid state below the solidus line and each domain of the mixture is made up of charge carries formed from the fragmentation of the bulk structure into its basic building units (IPs, ICs or solvent aggregates). The isothermal structural variation at high and low temperatures in a phase diagram with no solvate formation and in a phase diagram with a solvate formation are presented in Fig S.1 and S.2, respectively. As shown, the main features of the heterogeneous structure are (1) a molten solvent domain (Hydrogen bonded tetrahedral structure with tetrahedron building units in the case of water) and molten eutectic domain (water and IPs) below the eutectic composition ($x_{eutectic}$) which can be represented as a molten eutectic-in-water domain, (2) a molten eutectic domain only at the eutectic composition ($x_{eutectic}$) with a distinct melting point ($T_{eutectic}$) and a distinct structure of alternate layered molten solvent domain and molten salt domain, (dominated by un-solvated contact ion pairs (UCIP)) or solvate domain (dominated by solvated contact ion pairs (SCIPs and ICs)), (3) a mixture of molten eutectic domain as in (2) and a molten solvate domain (dominated by SCIPs and ICs) or molten salt domain (dominated by CIPs and ICs) if there is no solvate formation above $x_{eutectic}$, which can be presented as molten eutectic-in-molten salt domain or molten eutectic-in-molten solvate domain. It was emphasized that the building units or charge carriers are kinetic entities that undergo a rapid dynamic exchange among each other and that free volume and activation energy play a key role in controlling the transport specially when crossing $x_{eutectic}$ which shows the lowest activation energy for transport. This idea of liquid heterogeneity is not new and in fact mounting evidence of recent studies point to nanoheterogeneity in liquids[13]. It was even suggested that at low enough temperatures liquid-liquid separation could take place due to incompatibility between the two solvent-rich domains and solvate-rich or molten salt-rich domains[12]. Individual schematic illustrations of the salt/water structure at all possible molar fraction-temperature combinations for both kind of phase diagrams (with and without solvent formation) according to the model can also be found in the supplementary information section of this work (Fig S.3).

In this work, we use this model, which was previously applied to aqueous and non-aqueous solutions of alkali metal salts[8], in order to understand the structure and ionic conductivity of four selected nitrate-based electrolyte solutions. The four solutions were carefully selected so that they have diverse phase diagrams with and without solvate formation and diverse isotherms with a conductivity maximum that does and does not drop beyond the onset maximum molar fraction. Below is a brief description for the chemistries under study:

– Ammonium nitrate/water ($NH_4NO_3/H_2O$): This phase diagram has a simple eutectic and no solvate formation and a conductivity isotherm with a maximum conductivity that drops slightly thereafter.

– Lithium nitrate/water ($LiNO_3/H_2O$): This phase diagrams have a solvate forming and two eutectic points and their conductivity isotherms have a maximum conductivity that drops thereafter.

– Nitric acid/water ($HNO_3/H_2O$): This phase diagram has two solvate formation points and three eutectic points and a

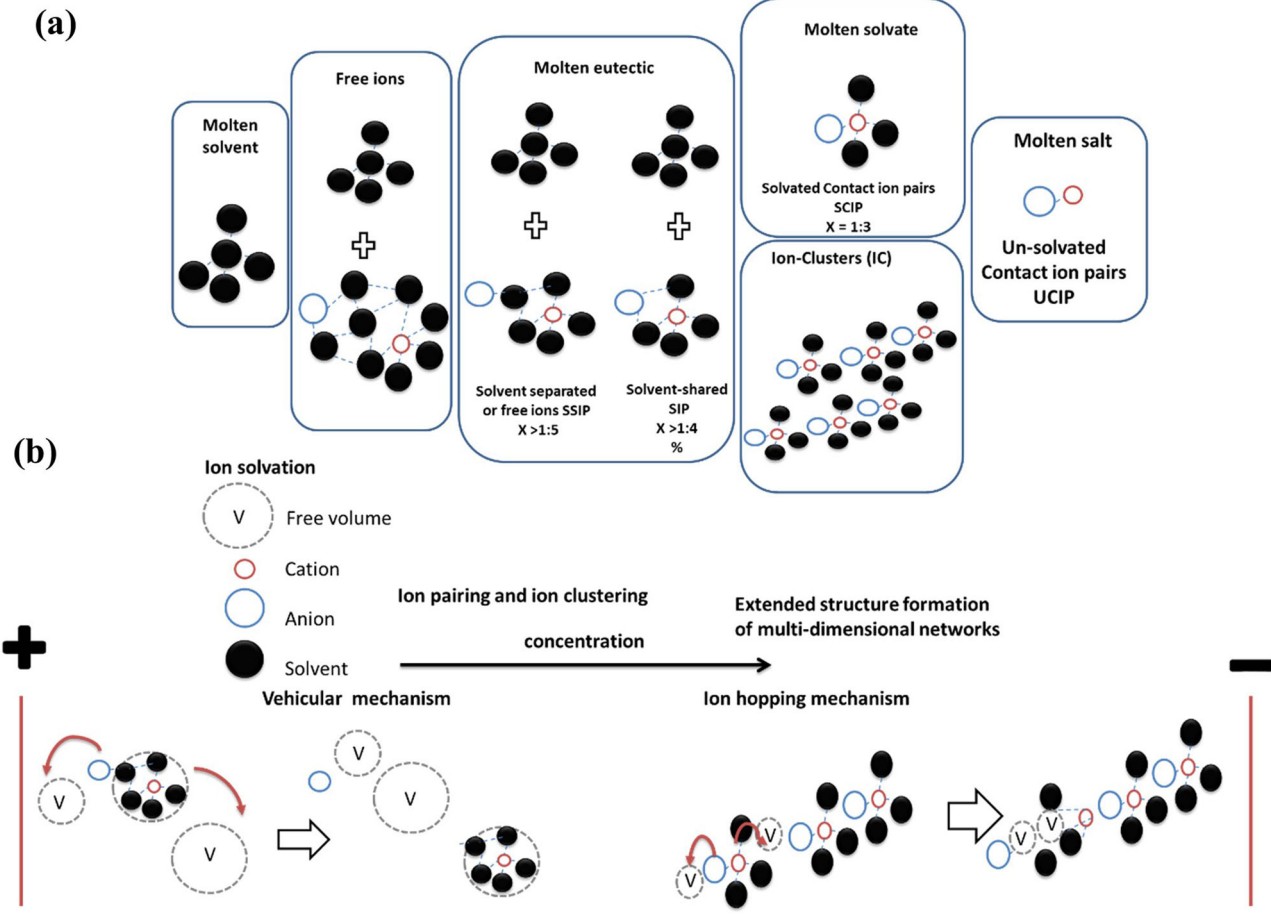

**Fig. 1 Schematic illustration of microstructural domains. a** Possible structures of the different IPs and ICs that can exist in electrolyte solutions at different concentrations. **b** Illustration of the two different conduction mechanisms at different concentrations.

conductivity isotherm with a maximum conductivity that drops thereafter.

- Calcium nitrate/water ($Ca(NO_3)_2/H_2O$): This phase diagram has 3 solvate forming with the latter 2 forming within a very narrow compositional range. For simplicity, we modified the phase diagram to include one solvate formation. This system becomes similar to that of Lithium nitrate/water.

First, we examine the conductivity isotherms and the phase diagrams and show that the correlations follow the model. Next, we use well-established one-variable, isothermal and iso-compositional equations to fit the data. We, then, propose a two-independent-variables equation that fits the data well as indicated by the coefficient of determination, $R^2$.

## Results and discussion

**Ionic conductivity of aqueous nitrate electrolyte solutions in relation to their phase diagrams.** Figure 2a–d shows plots of conductivity isotherms i.e., the specific ionic conductivity (κ), vs. molar fraction (x) at 298 K of the four aqueous electrolyte solutions along with their respective phase diagrams. In order to better understand the observed isotherms, let us have a closer look at the binary phase diagrams and extract as much information as we can about the structure of the liquid from those of the corresponding solids for a given composition at lower temperatures.

**The phase diagram with simple eutectic and no solvate formation: $NH_4NO_3/H_2O$.** Figure 2a shows the phase diagram which is a simple eutectic with no solvate formation. The melting point of the electrolyte solution decreases from 273.2 K($T_{m\ (H2O)}$) with increasing the amount of salt, as represented by the molar fraction x, until it reaches a minimum 257 K ($T_e$), at the eutectic molar fraction 0.14, then increases again up until that of the pure salt 443 K ($T_{m\ (NH4NO3)}$). The electrolyte solution is liquid above $T_m$ and liquid and solid between $T_m$ and $T_e$ and solid below $T_e$. In order to understand the structure of the liquid it is better to understand that of the solid. Below $T_e$ at $x_e$, the solid is made of pure eutectic while at molar fractions below $x_e$ it is made of a mixture of solid eutectic and pure solid solvent and above $x_e$ it is made of solid eutectic and pure salt. The structure of solid eutectics has been studied in many systems and more so in metal alloys[14]. It is clear that due to kinetic and thermodynamic limitations the eutectic tends to adopt certain peculiar structures where the two end compounds, in this case the solvent and the salt, tend to alternate into a lamellar structure in most systems. We can hypothesize that this structure is carried over upon melting to the liquid state. Therefore, at $x_{max}$ in the conductivity isotherm, which is essentially $x_e$, the alternate structure has the highest conductivity (the maximum of the product of mobility and charge carriers). Ongoing below or above this concentration a diluting effect occurs where the composition of eutectic in the mixture is reduced and replaced by the less conductive bulk solvent (molten eutectic-in-water or molten eutectic-in-molten salt domain). However, the drop in the lower end goes to approximately zero due to the poor conductivity of pure, bulk solvents while in the higher end does not drop to zero due to the more conductive molten salt. These drops in conductivity on both sides of the $x_{max}$ explains the observed maximum.

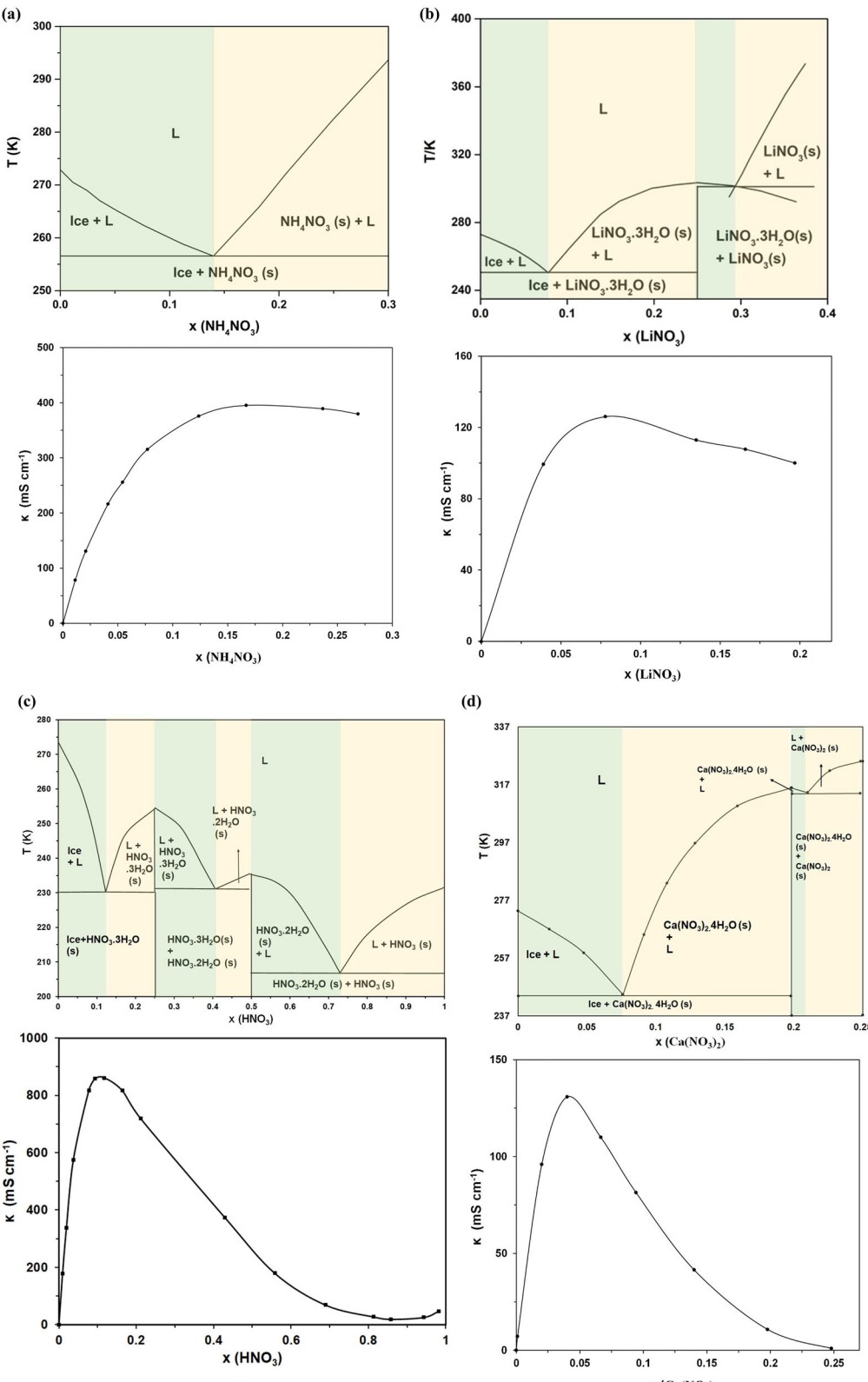

**Fig. 2 Phase diagram and room temperature ionic conductivity isotherms. a** Ammonium nitrate (**b**) lithium nitrate (**c**) nitric acid (**d**) calcium nitrate.

**The phase diagram with solvate formation: LiNO₃/H₂O and Ca(NO₃)₂/H₂O**. Figures 2b and 3d show the phase diagrams which illustrate a solvate formation. The following is a description of LiNO₃/H₂O phase diagram which share the same characteristics as Ca(NO₃)₂/H₂O phase diagram. Figure 2b shows a solvate

formation at $x = 0.25$ ($x_s$) molar fraction corresponding to LiNO₃.3H₂O. The phase diagram now is split into two simple eutectic phase diagrams like the one in Fig. 2a but on both sides of the LiNO₃.3H₂O solvate. i.e., H₂O/LiNO₃.3H₂O and LiNO₃.3H₂O/LiNO₃. The melting point of the electrolyte solution

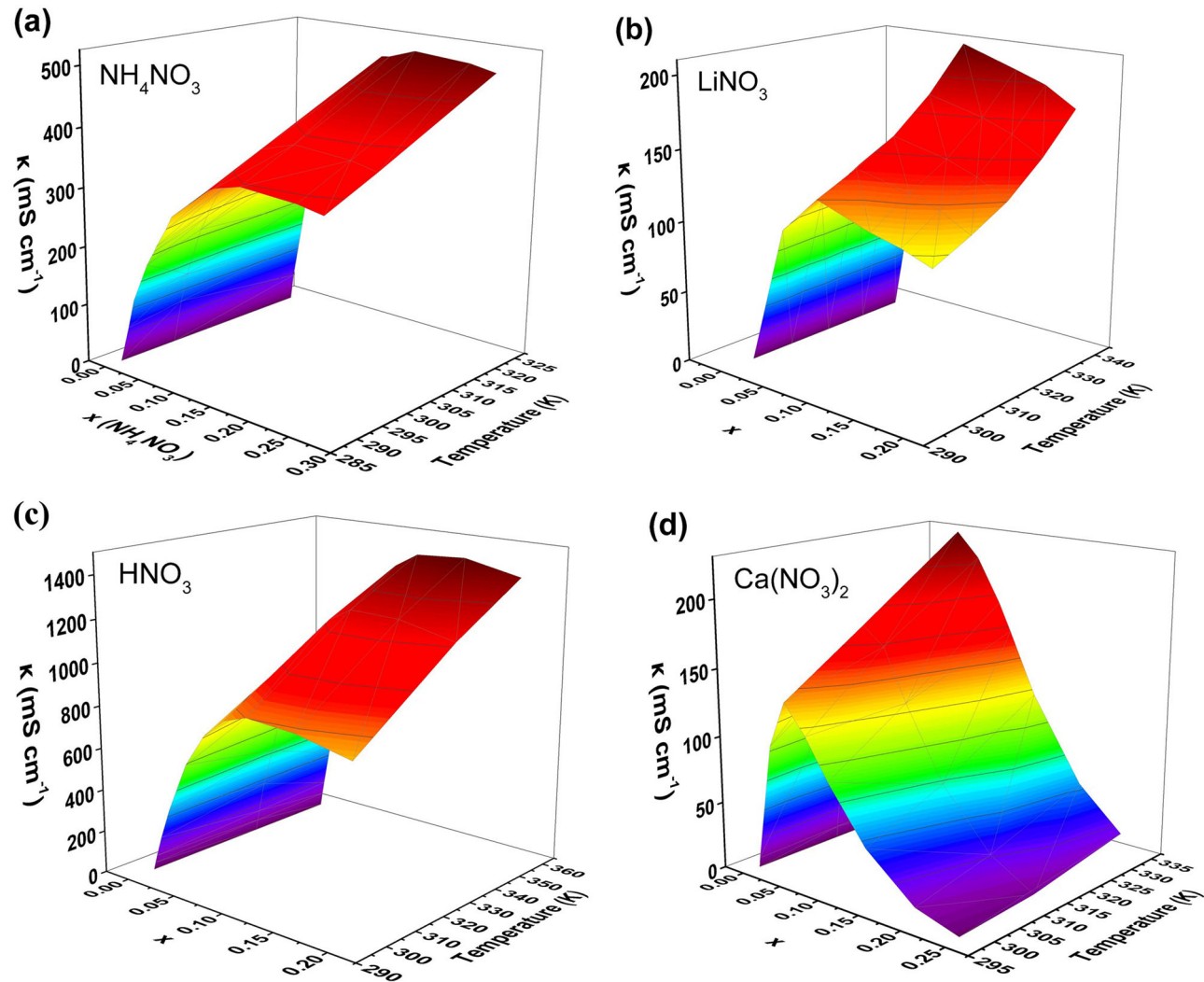

**Fig. 3 The variation of ionic conductivity (κ) with molar fraction (x) and temperature (T). a** Ammonium nitrate (**b**) lithium nitrate (**c**) nitric acid (**d**) calcium nitrate.

decreases from 273 K ($T_{m \ (H2O)}$) at x = 0 with increasing the amount of salt until it reaches 250 K ($T_{e1}$) at the first eutectic point ($x_{e1}$), then increases again up until that of the $LiNO_3.3H_2O$ solvate ($T_{ms}$) of 304 K, at $x_s$ of 0.25. $T_{e1}$ remains almost constant throughout this range at 304 K. Going higher in x, $T_m$ decreases again until it reaches 301 K at the second eutectic point ($T_{e2}$), then increases again up until that of the $LiNO_3$ salt at $T_{m2}$ of 526 K. $T_{e2}$ remains constant throughout this range at 301 K. In this case the electrolyte solution is liquid above $T_m$ and liquid and solid between $T_m$ and $T_{e1}$ and in between $T_m$ and $T_{e2}$ while solid below $T_{e1}$ and $T_{e2}$. The structure of the liquid resembles that of the solids below $T_{e1}$ and $T_{e2}$. At $x_{e1}$ and $x_{e2}$, the solids are made of pure eutectics while below $x_{e1}$ it is made of a mixture of solid eutectic and pure solid solvent while above $x_{e1}$ and below $x_s$ it is a mixture of a solid eutectic and a solvate. Between $x_s$ and $x_{e2}$, it is a mixture of a solid eutectic and a solvate. While above $x_{e2}$ it is a mixture of a solid eutectic and pure salt. Now, in this type of phase diagram the eutectic is made up of the solvent and the $LiNO_3.3H_2O$ solvate and shows the highest conductivity at $x_{max}$ in the conductivity isotherm. Also, ongoing below or above this concentration the structure changes from molten eutectic-in-water to molten eutectic-in-molten $LiNO_3.3H_2O$ solvate accompanied by a drop in conductivity at both ends but the lower end goes to approximately zero due to the poor conductivity of pure

water while in the higher end does not drop to zero due to the more conductive molten $LiNO_3.3H_2O$ solvate. The $Ca(NO_3)_2/H_2O$ phase diagram shows the same features but with a $Ca(NO_3)_2.4H_2O$ solvate forming at higher ($T_{ms}$) temperature than $LiNO_3.3H_2O$ solvate at 315 K and $x_s$ of 0.2. In this case however, the conductivity in the higher end drops to zero indicating lower conductivity of the molten $Ca(NO_3)_2.4H_2O$ solvate due to tis higher stability.

**The phase diagram with two solvate formation: $HNO_3/H_2O$.**
Figure 2c shows the phase diagram which illustrates two solvate formations at $x_{s1} = 0.25$ and $x_{s2} = 0.5$ corresponding to $HNO_3.3H_2O$ and $HNO_3.2H_2O$, respectively. The phase diagram is now split into three simple eutectic phase diagrams like the one in Fig. 2a. The first is to the left of the $HNO_3.3H_2O$ solvate i.e., $H_2O/HNO_3.3H_2O$ and the other two are on both sides of $HNO_3.2H_2O$ solvate i.e., $HNO_3.3H_2O/HNO_3.2H_2O$ and $HNO_3.2H_2O/HNO_3$. The melting point of the electrolyte solution decreases from 273 K ($T_{m \ (H2O)}$) at x = 0 with increasing amount of salt until it reaches 230 K ($T_{e1}$), then increases again up until that of the $HNO_3.3H_2O$ solvate ($T_{ms1}$) of 254 K at x = 0.25. $T_{e1}$ remains constant throughout this range at 230 K. between $x_{e1}$ and $x_{e2}$, $T_m$ decreases again until it reaches 231 K ($T_{e2}$), then increases again up until that of the $HNO_3.2H2O$ solvate $T_{ms2}$ of 235 K. $T_{e2}$

**Table 1 Lists the cation charge density, Gibbs free energy of interaction ($\Delta G^{int}$), eutectic molar fraction ($x_{eutectic}$) and temperatures ($T_{eutectic}$), maximum ionic conductivities ($\kappa_{max}$) and corresponding molar fractions ($x_{max}$), as well as the solvate formation molar fractions ($x_{solvate}$) for all the aqueous electrolyte solutions.**

| Solute | Chemical structure | Cation charge density (e $Å^{-3}$) | $\Delta G^{int}$ (kJ $mol^{-1}$) | $x_{eutectic}$ | $T_{eutectic}$ (K) | $x_{max}$ | $\kappa_{max}$ (mS $cm^{-1}$) | $x_{solvate}$ |
|---|---|---|---|---|---|---|---|---|
| Ammonium nitrate | $NH_4NO_3$ | 0.048[17] | −45.40[17] | 0.14 | 256 | 0.16 | 396 | – |
| Lithium nitrate | $LiNO_3$ | 0.327[17] | −115.8[17] | 0.08 | 250 | 0.08 | 126 | 0.25 |
| Nitric acid | $HNO_3^{*}$ | – | – | 0.12 | 230 | 0.12 | 860 | 0.25 |
|  |  |  |  | 0.41 | 231 |  |  | 0.5 |
|  |  |  |  | 0.73 | 206 |  |  |  |
| Calcium nitrate** | $Ca(NO_3)_2$ | 0.322[17] | −192.5[17] | 0.07 | 244 | 0.04 | 131 | 0.2 |
|  |  |  |  | 0.25 | 314 |  |  |  |

*$NO_3$: charge density = 0.015 e $Å^{-3}$ and interaction energy $\Delta G^{int}$ = −17.8770 kJ $mol^{-1}$,[17]. **only divalent solute; provides double cation molar fraction at any given salt molar fraction.

remains constant throughout this range at 231 K. beyond $x_{e2}$, $T_m$ decreases again until it reaches 207 K ($T_{e3}$), then increases again up until that of pure $HNO_3$ salt of 231 K ($T_{m\ (HNO3)}$). $T_{e3}$ remains constant throughout this range at 207 K. In this case, the electrolyte solution is liquid above $T_m$ and liquid and solid between $T_m$ and $T_{e1}$, between $T_m$ and $T_{e2}$, and in between $T_m$ and $T_{e3}$ while solid below $T_{e1}$, $T_{e2}$, and $T_{e3}$. The structure of the liquid resembles that of the solids below $T_{e1}$, $T_{e2}$, and $T_{e3}$. At the eutectic molar fractions, the solids are made of pure eutectics while below $x_{e1}$ it is made of a mixture of solid eutectic and pure solid solvent while above $x_{e1}$ and below $x_{s1}$ it is a mixture of a solid eutectic and the $HNO_3.3H_2O$ solvate. Between $x_{s1}$ and $x_{e2}$ it is a mixture of a solid eutectic and $HNO_3.3H_2O$ solvate. While above $x_{e2}$ and below $x_{s2}$, it is a mixture of a solid eutectic and $HNO_3.2H_2O$ solvate. Similarly, above $x_{s2}$ and below $x_{e3}$, it is a mixture of a solid eutectic and $HNO_3.2H_2O$ solvate while above $x_{e3}$, it is a mixture of a solid eutectic and pure $HNO_3$ salt. Similar to $LiNO_3/H_2O$ and $Ca(NO_3)_2/H_2O$ phase diagrams the eutectic is made up of the solvent and the $HNO_3.3H_2O$ solvate and shows the highest conductivity at $x_{max}$ in the conductivity isotherm. Also, ongoing below or above this concentration the structure changes from molten eutectic-in-water to molten eutectic-in-molten $HNO_3.3H_2O$ solvate then to molten eutectic-in-molten $HNO_3.2H_2O$ solvate. However, in this case the drop in conductivity at both ends goes to approximately zero due to the poor conductivity of pure water and pure $HNO_3$.

Generally, the first eutectic point ($x_{eutectic}$) for each of the solutions typically corresponds to the molar fraction with the highest ionic conductivity ($x_{max}$). This can be seen clearly in the case of ammonium nitrate, lithium nitrate and nitric acid. In the case of calcium nitrate, $x_{max}$ lies slightly after $x_{eutectic}$. This behavior is typically witnessed only for the first eutectic point (at lower concentration of salts) and is not observed for subsequent eutectic points. One exception to this observed behavior is exhibited by sulfuric acid in water system where a second peak is seen in the conductivity isotherms at subsequent eutectic point that corresponds to its fourth eutectic point at x = 0.71[15,16]. For most of the studied electrolyte solutions, it is found that subsequent eutectic points and solvate formation points do not have a unique common effect on the room temperature conductivity isotherm. Table 1 gathers the information about the cation in the nitrate salts studied, eutectic compositions and corresponding conductivity at these compositions, eutectic temperatures and the solvate compositions.

From Table 1, we can see that the ionic conductivity of all the aqueous electrolyte solutions ($\kappa_{max}$) follows the order: $H^+$ (860 mS $cm^{-1}$) > $NH_4^+$ (396 mS $cm^{-1}$) > $Ca^{2+}$ (131 mS $cm^{-1}$) > $Li^+$ (126 mS $cm^{-1}$). This could be understood based on the charge density of the cations as shown in Table 1. The higher the charge

density (e.g., Li and Ca) the stronger the interactions with water molecules as evidenced by the interaction energy ($\Delta G^{int}$) as shown in Table 1. Also, the stronger the interaction the higher the activation energy and the lower the mobilities of the cations. We here assumed that the anion has little interaction with water as evidenced by the low $\Delta G^{int}$ of the nitrate anion which reported to be −17.88 kJ $mol^{-1}$,[17]. For both $x_{max}$ and $x_{eutectic}$ the trend is the same as follows: $NH_4^+$ > $H^+$ > $Li^+$ > $Ca^{2+}$. This corroborates the correlation between the two parameters. The trend however for the position of $x_{max}$ and $x_{eutectic}$ is still not understood. Empirically it is found that solutions of salt of higher valence cations and anions show lower $x_{max}$ and $x_{eutectic}$[3]. Possible explanation is again that the higher charge density of the cation and/or the anion attracts more solvent molecules from the water layer into the molten salt or solvate layer hence increasing the number of charge carriers (IPs and ICs). This causes a faster drop in the ratio of ice to eutectic shifting x to the left of the phase diagram. Similar line of thought can be applied to $T_{eutectic}$. Herein the trend is $NH_4^+$ > $Li^+$ > $Ca^{2+}$ > $H^+$ which again can be explained by differences in their charge density as faster break up in water bulk structure and larger number of IPs leads to lower $T_{eutectic}$. It is worth mentioning that proton conductivity in acids is given special attention and it is believed that the very high proton conductivity is due to Grotthuss mechanism where the proton can hop into the water network and contribute to the total conductivity along with the vehicular mechanism of the protonated water molecules. i.e. both mechanisms are at play[18]. It is also worth noting that the water-molten salt eutectic of $NH_4^+$ has higher conductivity than that of the water-solvate of $Li^+$ and $Ca^{2+}$. This implies that the UCIPs and possibly ICs are more dissociated by water molecules (or more mobile) than the more stable SCIPs.

Figure 3 illustrates the ionic conductivity behavior vs. molar fraction of the nitrate salts in the aqueous electrolyte solutions at different temperatures. Notably, there is no shared trend in behavior across the four salts other than the observation that $\kappa_{max}$ increases with temperature. Nitric acid has the highest $\kappa$ reaching well above 1 S $cm^{-1}$ around its $x_{max}$ at temperatures higher than 60 °C. Figure 3c shows a small plateau in $\kappa$ values around $x_{max}$ (x = 0.08 – 0.12) before they drop down again at x > 0.12. As mentioned above, the conductivity in acid solutions needs more attention as the very high conductivity is always explained by a combination of vehicular and ion hopping mechanisms taking place concurrently in solution. The electrolyte with the second highest $\kappa$ is ammonium nitrate, with values reaching up to 0.5 S $cm^{-1}$ around its $x_{max}$ at temperatures higher than 50 °C. The plateau behavior is once again observed but this time it persists at all compositions with x > $x_{max}$. This is observed because the solution structure is dominated by UCIPs of the

molten $NH_4NO_3$ originating from both the molten eutectic ($NH_4NO_3/H_2O$) domain, and molten salt ($NH_4NO_3$) domain. This is corroborated by the phase diagram which is a simple eutectic with no formation of a solvate. Lithium nitrate and calcium nitrate are the electrolytes with the least ionic conductivity reaching a maximum of $0.2\,S\,cm^{-1}$ around their $x_{max}$ at temperatures above 60 °C. Both electrolytes do not show any plateau in $\kappa$ values beyond their $x_{max}$. Rather, their ionic conductivity keeps decreasing with increasing x after $x_{max}$. Again, this can be explained by the dilution effect when the structure switches from molten eutectic-in water to molten eutectic-in-molten solvate. In this case the molten solvate (dominated by SCIPs) being less conductive than the molten salt (dominated by UCIPs) due to the high energy density of the cations.

Analysing the iso-compositional trends, we can see the conductivity of the solution rises with increasing temperature with varying proportionality as a function of x. In the dilute region, the increase in temperature results in subtle increase in conductivity while in the more concentrated region near solubility, temperature change results in a substantial increase in conductivity. This is mainly because at the high concentration regions, the number of ion pairs and clusters (IPs and ICs) is larger than that at the dilute region, and the ion association decreases with rising temperature, leading to more mobile "free" ions. Hence, the conductivity of the high concentration region is more temperature-sensitive[8,19]. Also, temperature expands the liquid and increases its free volume and lowers activation energy hence improving ion mobility.

**Ionic conductivity model description.** In the low concentration regions, $\kappa$ of the aqueous electrolyte solutions is equal to the total of the conductivities of the ions in the solution, according to the equation below[20].

$$\kappa = \sum n_i q_i \mu_i \qquad (1)$$

where $\kappa$ is conductivity, $q_i$ is the charge of the ion, $n_i$ is the number of ions and $\mu_i$ is the mobility of the ions. It can be demonstrated in Eq. (1) that the conductivity is impacted by both the quantity of ions in the solution and their mobility. To further expand the validity of the equation to medium and high concentrations, Eq. (1) must be adjusted to account for the effect of electrolyte concentration on the number of free ions and, accordingly, ion mobility.

In solutions of moderate and high concentrations, the distance between anions and cations decreases, and less mobile or non-conductive ion pairs are created by ion association, resulting in a reduction in the number of free ions involved in vehicular conduction. Hence, the quantity of free ions as a function of electrolyte concentration follow a linear relationship[19]. The following equation may be used to explain the number of free ions in relation to the molar fraction x:

$$n = ax \qquad (2)$$

where a is a constant and n is effectively the speed of ions per unit electric field intensity and is the product of the combined effects of the external electric field force and ion movement resistance. Considering the ion and its hydration layer as a whole, the resistance to movement during migration when an external electric field is applied consists of ion-ion, ion-solvent, and solvent-solvent forces. The first is a long-range interaction induced by electrostatic forces, while the second and third are short-range interactions. At low concentrations, long-range interactions dominate while short-range interactions are often neglected. When the electrolyte concentration in the solution rises, the distance between molecules drops, and it becomes

**Table 2 Fitted parameters for Eq. (4) for different systems at different temperatures.**

| System | T | Parameters | | | Data source |
|---|---|---|---|---|---|
| | | A | B | $R^2$ | |
| $NH_4NO_3$ | 288.15 | 5394 | 5.515 | 0.9963 | 24 |
| | 298.15 | 6355 | 5.720 | 0.9947 | |
| | 323.15 | 8496 | 5.950 | 0.987 | |
| $LiNO_3$ | 298.15 | 3642 | 10.38 | 0.9865 | This work |
| | 308.15 | 4067 | 10.26 | 0.9903 | |
| | 318.15 | 4622 | 10.35 | 0.9847 | |
| | 328.15 | 5066 | 9.99 | 0.9895 | |
| | 338.15 | 5948 | 9.99 | 0.9868 | |
| $HNO_3$ | 298.15 | 23,290 | 10.11 | 0.9976 | 25 |
| | 333.15 | 32,430 | 9.777 | 0.9984 | |
| | 358.15 | 37,270 | 9.595 | 0.999 | |
| $Ca(NO_3)_2$ | 298.15 | 8143 | 23.84 | 0.9958 | 3 |
| | 313.15 | 10,770 | 22.56 | 0.9987 | |
| | 333.15 | 10,443 | 16.41 | 0.9377 | |

harder to neglect short-range interactions; hence, ion migration resistance rises quickly[21]. Therefore, ion mobility often decreases as electrolyte concentration rises. Zhang et al[19]. have proposed an empirical equation to relate $\mu$ and x that can be applied to electrolytic solutions in any concentration region:

$$\mu_i = \mu_{io} \exp(-bx) \qquad (3)$$

where b is a constant.

Substituting Eqs. (2) and (3) into Eq. (1), we arrive to the following equation relating $\kappa$ and x which was introduced before by Abu-Lebdeh et al.[8]:

$$\kappa = A\exp(-Bx) \qquad (4)$$

where A and B are constants. To test the validity of Eq. (4) on the different aqueous electrolyte system, we have fitted the data obtained from literature and our own experimental conductivity data for $LiNO_3$ to the equation. The parameters obtained and the $R^2$ values are tabulated in Table 2 below. $R^2$ column indicates the model fits the data well with $R^2 > 0.98$ for all the systems at different temperatures except for $Ca(NO_3)_2$ at the highest temperature (333 K; $R^2 \approx 94\%$).

While Eq. (4) represent the effect of electrolyte concentration on conductivity, it can only be applied to the study of different electrolyte solutions isothermally as conductivity also depends on temperature as illustrated in the famous Arrhenius equation:

$$\kappa = A_o \exp\left(\frac{-E_a}{RT}\right) \qquad (5)$$

Or also by the Vogel–Fulcher–Tammann (VTF) equation:

$$\kappa = \frac{A_o}{\sqrt{T}} \exp\left(\frac{-E_a}{R(T - T0)}\right) \qquad (6)$$

where $A_o$ is the pre-exponential factor, $E_a$ is the activation energy R is the gas constant and T is the temperature in Kelvin and $T_o$ is an empirical temperature related to $T_g$, the glass transition temperature.

Figure 4 presents the ionic conductivity variation of the four electrolyte solutions on a logarithmic scale vs. the reciprocal of the temperature. The temperature dependence of $\ln \kappa$ shows a similar linear trend across all electrolytes indicating that they all obey the classical Arrhenius behavior given by Eq. (5).

Using Fig. 4, we calculated the activation energies and the pre-exponential factor for each of the nitrates aqueous electrolytes and plotted each against the molar fraction, x.

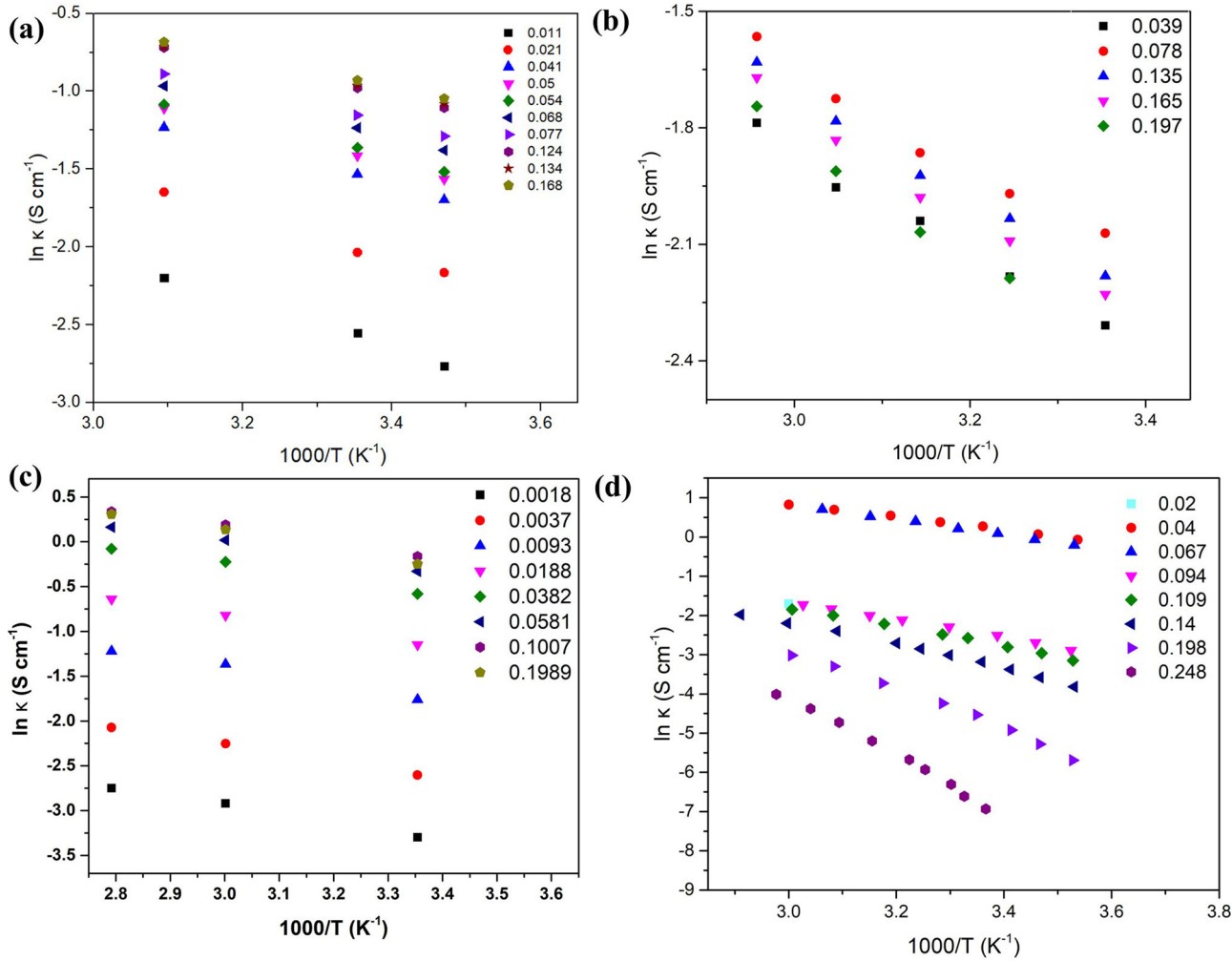

**Fig. 4 Ionic conductivity, ln (κ), of the different nitrate solutions as a function of the reciprocal temperature. a** Ammonium nitrate (**b**) lithium nitrate (**c**) nitric acid (**d**) calcium nitrate.

Figure 5 depicts the change in activation energy as the electrolyte molar fraction in solution (x) increase. The general trend observed is a decrease in activation energy as we approach $x_{eutectic}$ which is the expected behavior that accompanies the increase in ionic conductivity. This is then followed by an increase in the activation energy as we pass the eutectic point (x > $x_{eutectic}$) in all the electrolytes except ammonium nitrate where the activation energy plateaus beyond $x_{eutectic}$. This explains the plateauing of ionic conductivity of ammonium nitrate beyond $x_{eutectic}$ as observed in Fig. 2a. In addition, we have found that nitric acid has the least activation energy at dilute concentration (x < 0.01), followed by ammonium nitrate and lastly calcium nitrate (we do not have the dilute activation energy data for lithium nitrate). This observed trend indicates that electrolytes with highest conductivity tend to have the lowest activation energy at low x values. We therefore expect lithium nitrate to have a similar activation energy value at lower x values as calcium nitrate because of their similar ionic conductivity. The increase in the activation energy can be simply attributed to the higher number of the ion-ion interactions at the expense of solvent-solvent and ion-solvent interactions as the structure changes in the domains from molten eutectic-in water to molten eutectic-in molten solvate or molten eutectic-in-molten salt. Also, a drop in the free volume can also contribute to this behavior.

The intercept of Fig. 4, or the pre-exponential factor (A) was calculated and plotted against molar fraction (x) as shown in

Fig. 6. Once again, not all nitrates share the same trend. For instance, Ammonium nitrate's A decreased with an increase in molar fraction until it hit a plateau at around x = 0.2 While all the other electrolytes' A increased with an increase in molar fraction. Lithium nitrate A increased gradually at first before rising up significantly after x = 0.2 while nitric acid and calcium nitrate's A increased significantly at dilute solutions until x = 0.04 and then continued to ramp gradually after that concentration. The continuous increase of $E_a$ and A after $x_{eutectic}$ can explain the drop in ionic conductivity beyond that ratio. Upon observing ammonium nitrate's unique $E_a$ and A behavior, one can understand the reason behind the stagnation of ionic conductivity at $κ_{max}$. For the rest of the salt solutions, both activation energy and pre-exponential factor increase significantly at high concentrations and might be responsible for the severe drop in conductivity at high concentrations beyond $x_{max}$ and eutectic composition.

The degree of ion dissociation in the solution and the number of free ions both increase as the temperature rises.

$$n = a'T \qquad (7)$$

Furthermore, when the temperature rises, the intermolecular force reduces, which causes the barrier to ion movement to

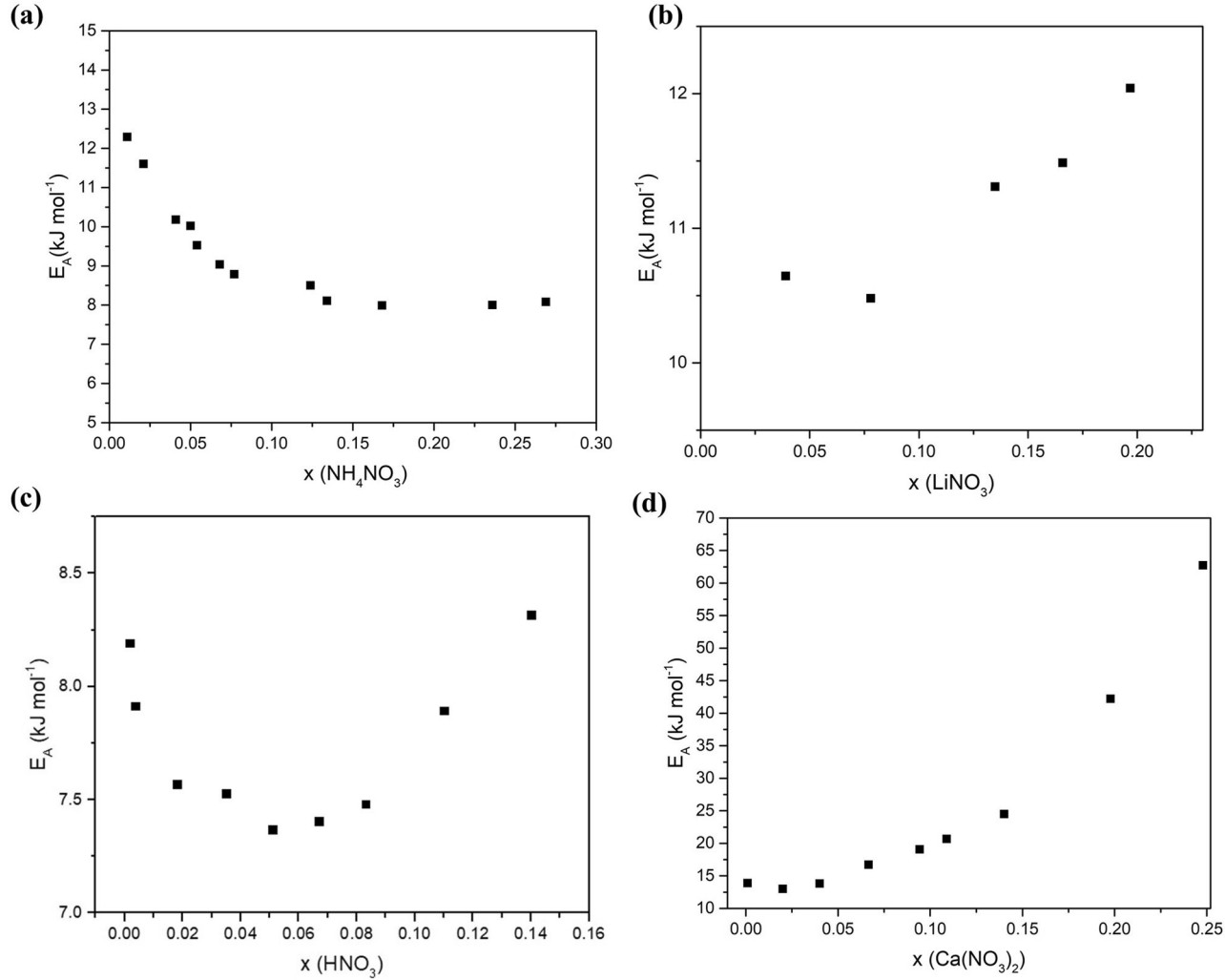

**Fig. 5 Activation energy for the different nitrate solutions as a function of molar fraction. a** Ammonium nitrate (**b**) lithium nitrate (**c**) nitric acid (**d**) calcium nitrate.

**Table 3 Fitted parameters for three possible κ vs. T empirical equations at different concentration regions.**

| Model | x region | Parameters | | |
| --- | --- | --- | --- | --- |
| | | A | B | $R^2$ |
| $\kappa = \mathbf{AT}\exp\left(\frac{-\mathbf{B}}{\mathbf{T}}\right)$ | Dilute | 8.967 | 984.5 | 0.9893 |
| | Near $x_{max}$ | 11.55 | 996.6 | 0.9823 |
| | Concentrated | 14.38 | 1105 | 0.9914 |
| $\kappa = \frac{\mathbf{A}}{\sqrt{\mathbf{T}}}\exp\left(\frac{-\mathbf{B}}{\mathbf{T}}\right)$ | Dilute | 229186 | 1463 | 0.9885 |
| | Near $x_{max}$ | 295284 | 1475 | 0.9805 |
| | Concentrated | 367972 | 1583 | 0.9903 |
| $\kappa = \mathbf{A}\sqrt{\mathbf{T}}\exp(-\mathbf{B})$ | Dilute | 264.1 | 1143 | 0.9890 |
| | Near $x_{max}$ | 340.2 | 1156 | 0.9817 |
| | Concentrated | 423.8 | 1264 | 0.9910 |

decrease and thus, ion migration speeds up:

$$\mu_i = \mu_{io}\exp\left(\frac{-b'}{T}\right) \qquad (8)$$

The dependance of ionic conductivity on temperature in the pre-exponential term could take many forms. Herein, we tested three different empirical equations listed in Table 3 alongside their calculated parameters and the $R^2$ values when fitted against LiNO$_3$ (κ vs.) T data.

It can be observed that there is only a slight advantage for the first linear model when comparing $R^2$. For this slight edge and the sake of simplicity, the linear dependence will be used in this work:

$$\kappa = AT\exp\left(\frac{-B}{T}\right) \qquad (9)$$

where A and B are constants.

Following the observed correlation of κ with x and T in Eqs. (4) and (9), we propose the following semiempirical equation:

$$\kappa = ATx\exp\left(\frac{-Bx}{T}\right) \qquad (10)$$

Note that this is not the first time such a model is proposed. Lin et al.[22] have analyzed the conductivity data for the ionic liquids [EMIM][C$_2$N$_3$] and [EMIM][CF$_3$SO$_3$] in aqueous solutions and developed a six-parameter empirical model, as shown in Eq. (11), that concurrently connected conductivity, electrolyte concentration, and temperature:

$$\kappa = x\exp(A_1+A_2T^{0.5}+A_3x^{0.5})+A_4+A_5T^{0.5}+A_6x^{0.5} \qquad (11)$$

where $A_1 - A_6$ are empirical parameters that can be obtained through data regression. Similarly, Eq. (8) is a seven-parameter model developed by Fu et al.[23] that is used to correlate ionic

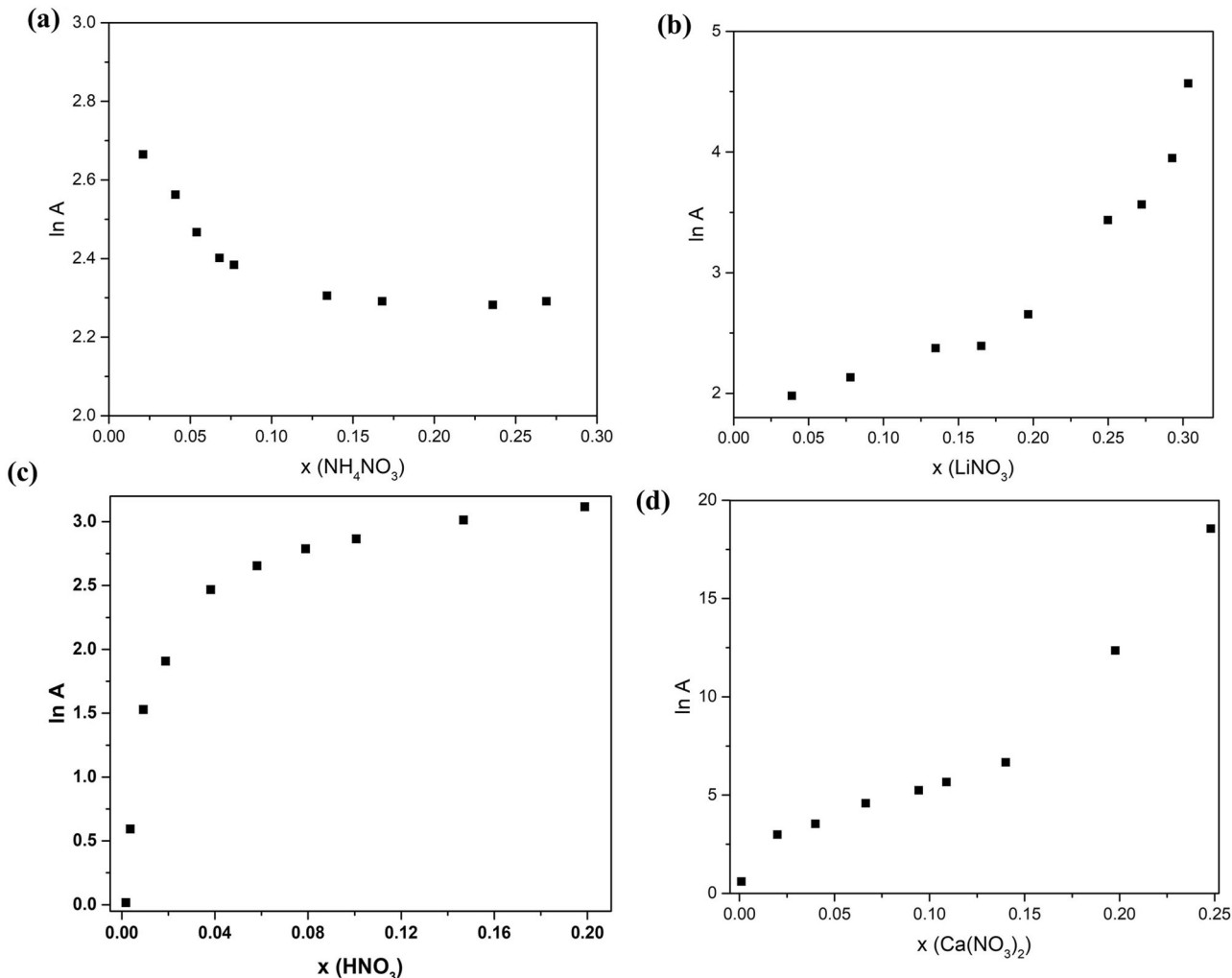

**Fig. 6 Pre-exponential factor for the different nitrate solutions as a function of molar fraction. a** Ammonium nitrate (**b**) lithium nitrate (**c**) nitric acid (**d**) calcium nitrate.

conductivity of ionic liquids in organic solvents:

$$\kappa = B_1 x^{B2} \exp\left(-\frac{B_4 x^{0.5} + B_5 x + (B_6 x' + B_7)^2}{T - B_3}\right) \quad (12)$$

where $B_1 - B_7$ are also empirical parameters and x' is a parameter related to solvent composition. Further, Zhang et al.[19] have proposed a five-parameter model that correlates the same components and is suitable for both pure and mixed solvent systems premised on how the electrolyte concentration and temperature affect the quantity and mobility of free ions:

$$\kappa = (P_1 T + P_2) m^n \exp\left(-\frac{P_3 m}{T - P_4}\right) \quad (13)$$

where $P_1 - P_4$ are also empirical parameters and n is a parameter related to solvent species. All of the mentioned models do a great job when fitting data with great degree of accuracy. The $R^2$ value for all the aforementioned studies is above 99%. Our model exhibited slightly less yet accepted accuracy with $R^2$ values always exceeding 95%.

Figure 7 presents the 3D graphs representing the collected data from literature and our experimental work (colored curves) next to the fitted model data (light blue curves). The parameters A, B and $R^2$ values for each of the nitrate aqueous electrolytes in this study are labeled on their respective 3D graphs. It can be clearly observed that the light blue curve underestimates κ at lower

temperature at most molar fractions and it overestimates κ at high temperature at all molar fractions. We acknowledge that the oversimplification of the problem into a two-parameter equation could be the main cause for the over/under estimation observed. However, despite the discrepancies having two parameters only offers several advantages that contribute to its practical utility. First, the simplicity of the model makes it more accessible and easier to interpret than models with a higher number of parameters. This ease of understanding is particularly valuable for researchers and practitioners who require a straightforward yet effective tool for predicting conductivity in various applications. Second, the lower number of parameters in our model decreases the risk of overfitting, ensuring that it is more likely to generalize well to new, unseen data. This is an essential attribute for a model that aims to be applicable across a wide range of electrolytes.

## Conclusions

In this work, we have studied the ionic conductivity of nitrate-salts/$H_2O$ aqueous solutions over wide concentration and temperature ranges and correlated the observed trends to their respective liquid-solid phase diagrams. We showed that the structure of the electrolyte solutions can be connected to the binary phase diagram in many ways: First, the $x_{max}$ in the conductivity vs. molar fraction plot coincides with the near eutectic

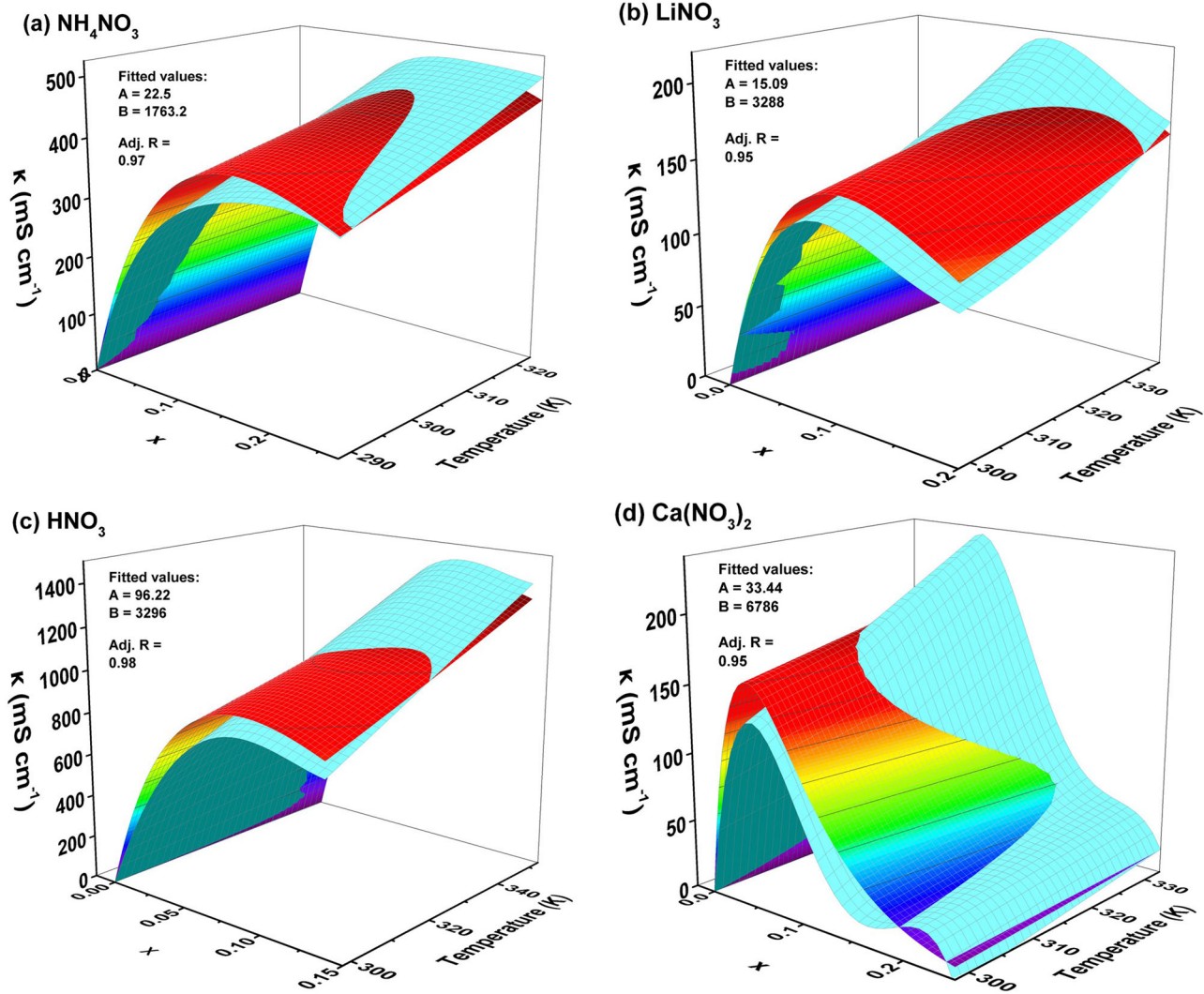

**Fig. 7 3D graphs of κ vs.** T and x for each of the nitrate systems plotted next to the fitted curve of Eq. (10). **a** ammonium nitrate (**b**) lithium nitrate (**c**) nitric acid (**d**) calcium nitrate.

composition. Second, the heterogeneous structure of the solid in the phase diagram persists in the liquid. This implies that upon increasing the concentration of the salt in the electrolyte solution, a transition in solution structure from molten eutectic-in water domain to molten eutectic-in molten solvate or molten eutectic-in-molten salt domains takes place. The activation energy was calculated from the Arrhenius equations and showed a minimum around $x_{max}$ and increased significantly thereafter. This was interpreted as the cause of the maxima in the conductivity isotherms and explained by the higher number of the ion-ion interactions at the expense of solvent-solvent and ion-solvent interactions and possibly changes in free volume. We also proposed a semiempirical model to correlate conductivity with molar fraction and temperature; $κ = f(x, T)$ and verified its accuracy vs. a wide concentration and temperature ranges. When compared to other conductivity models reported in the literature, ours proved to be of acceptable accuracy with fewer parameters; two vs. at least four reported by others. While our model may not achieve the same level of accuracy as more complex models with $R^2$ values above 99%, it still provides a reasonable approximation with $R^2$ values consistently above 95%. In many practical applications, this level of accuracy may be sufficient, particularly when considering the trade-offs in terms of simplicity, computational efficiency, and reduced overfitting risk.

**Experimental**. The ionic conductivity data of $NH_4NO_3$, $HNO_3$, $Ca(NO_3)_2$ solutions were obtained from Wahab et al.[24], Spencer et al.[25] and Angel et al.[3], respectively. $NH_4NO_3$ data was obtained in S m$^{-1}$ through the digitization of the plots using Automeris software and converted to mS cm$^{-1}$. $HNO_3$ data was directly taken from tables and required no digitization or conversion. For $Ca(NO_3)_2$, Data was digitized and converted from equivalent conductance units ($Ω^{-1}$ cm$^{-1}$ eq$^{-1}$) into mS cm$^{-1}$ data points. In our search for $LiNO_3$ data, we found conductivity plots reported by Li et al.[26] that exhibited two peaks in conductivity at different molar fractions, specifically $x = 0.11$ and $x = 0.19$. In this set of data specifically the second peak in the conductivity isotherm does not correlate to the phase diagram given that it is at a molar fraction between a eutectic point and solvate forming at $x = 0.25$. We hypothesize that a drop in conductivity must occur after the first peak. For this reason, we decided to run our own experiment to obtain $LiNO_3$ conductivity vs.. x and T data. $LiNO_3$ electrolyte solutions were prepared by dissolving known amounts of $LiNO_3$ salt (Aldrich) in de-ionized water, represented in salt to solvent molar fractions (x), and left to equilibrate overnight. Ionic conductivity measurements were carried out using conductivity meter RL060C from Thermo Scientific equipped with a water bath to control the temperature between 20 and 80 °C. The estimated error in measurement is ±1%. The variation between

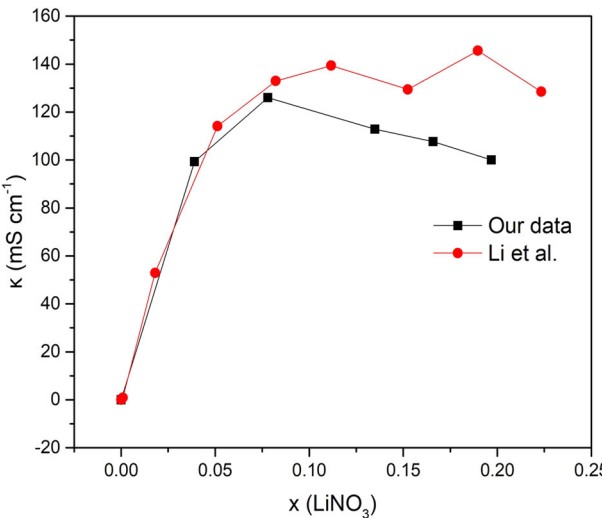

**Fig. 8 LiNO₃ conductivity vs. molar fraction data.** comparison of our LiNO₃ conductivity data with Li et al. data[26].

our data and Li et al data is presented in Fig. 8. The phase diagrams were obtained by direct digitization from literature[3,27–30].

## Data availability

The data that support the findings of this study are available from the corresponding author on reasonable request.

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

## Acknowledgements
We would like to thank Natural Resources Canada for partial financial support through the Program for Energy Research and Development (PERD).

## Author contributions
H.A.-S. has done data processing and analysis and developed Figs. 2–8, all under guidance of Y.A.-L. H.A.-S. also did the writing. Figure 1 was made by Y.A.-L.. Revision and editing were done by Y.A.-L. and E.A.B.

## Competing interests
The authors declare no competing interests.
