## [Peer Review File · Communications Chemistry]

Reviewers' comments:

Reviewer #1 (Remarks to the Author):

In this manuscript, the authors applied nitrate salts/H₂O system as a model to study the ionic conductivity variation by concentration and temperature. Based on the data from previous reports, the authors proposed a new model to emphasize this relationship, where the accuracy is over 95%. However, the authors did not clarify the relationship between ionic conductivity variation and solvation structure difference. Furthermore, previous researchers have studied similar models, but the authors did not differentiate their model from others', e.g., accuracy, benefits, etc. Therefore, the paper can not be considered for publishing in CommsChem until the author appropriately addresses the following issues::

- 1) The authors selected three different types of eutectic systems to study the ionic conductivity of nitrate solutions. However, all the K_{max} is at the eutectic composition and the conductivity decreases afterward. Therefore, will the number of eutectic points and solvation structures affect this trend?
- 2) Does all the experimental data (except LiNO₃) come from previous reports? Did the authors get the original data and how accurate are they?
- 3) At eutectic points, all the systems demonstrate the highest conductivity and the trend is H to Ca. As the authors claimed, it is due to the charge density of cations. However, protons use a Grotthuss mechanism to transport instead of a Viechle mechanism. Therefore, are they compatible?
- 4) The author assumed that the anion would not affect the conductivity. Is there any evidence? Moreover, the concentration of the cation varies a lot in different systems. Will the cation concentration affect the conductivity? At the same concentration (dilute), will the ionic conductivities follow this trend?
- 5) For the modeling part, most of the content and equations are similar to previous papers, such as ACS Omega 2020, 5, 22465–22474. The authors applied the Arrhenius equation to bring in the temperature factor. What is the benefit of this model? Are they different from previous reports? It seems like combining several constants to A and B. The authors should provide more information.
- 6) Does this equation fit all systems? Unfortunately, the Ca one does not fit very well. Also, except nitrate system, will changing anions influence this model?
- 7) The authors should enrich the description for the figure captions. They are not very clear. Additionally, from Figure 3 to Figure 7, they are mismatched with the content in the main text. Please correct them.
- 8) Please carefully check the paper. Multiple typos, grammar, and reference issues.

Reviewer #2 (Remarks to the Author):

In the present work, Al-Salih and co-workers investigated the transport behavior of aqueous solution containing different cations, and attempted to connect the structural changes in the phase diagrams with ionic conductivity. The results are quite interesting and the manuscript is well organized, I would therefore recommend the publication of this work after addressing the following minor issues.

1. Figure 1. To improve the readability of this figure, it is recommended to further illustrate the concepts of vehicular and ion-hopping mechanisms.
2. Line 142. The figure presented does not follow the numeric order. Please revise it.
3. Lines 148-152. It is not clear why the conductivity values reported by Li et al. differ from those reported in this work.
4. The transport of ionic species is tightly associated with the ligand exchange rate (i.e., Eigen value), as discussed in recent work (Energy & Environmental Science 2023, 16, 11). The authors are recommended to include this point in the revised version.

Reviewer #3 (Remarks to the Author):

Al-Salih et al. have studied the ionic conductivity (κ) as a function of molar ratio (x) and Temperature (T) for aqueous solutions of salts with nitrate anion and different cations (proton, lithium, calcium, and ammonium) along with their liquid-solid phase diagrams by Semi-Empirical Modeling. The idea is interesting, but some problems are still confusing. Questions are shown as following:

1. The authors propose four systems such as Ammonium nitrate/water and Lithium nitrate/water etc. The authors think that Ammonium nitrate/water system has a simple eutectic and no solvate formation and a conductivity isotherm with a maximum conductivity that drops slightly thereafter etc. However, I wonder how the authors ensure these systems have their proposal properties.
2. The authors use their own tested data for LiNO_3 , but other data are obtained from the references, such as NH_4NO_3 , HNO_3 , $\text{Ca}(\text{NO}_3)_2$. I really worry that these data can not support their proposal ideas. The authors should better use their own data.
3. Some illustrations don't match the Figures. For example, Fig 3a-d should be Figure 2a-d in page 6. The authors should check the details throughout their manuscript.
4. I confuse how the authors obtain the phase diagram in Figure 2, through calculation or experiments? The specific details should be provided.
5. The authors think that their formular can apply to high concentration solutions, however, the molar ratio is less than 0.5 as shown in Figure 5 and 6. It should be noticed that the realistic concentration of electrolyte is very high ($> 1 \text{ mol/L}$) in batteries, so I wonder how the authors think about higher concentration of electrolyte.

RE: manuscript ID: COMMSCHEM-23-0228

July 10, 2023

Dear Reviewers

Thank you for handling our manuscript. We are very grateful for your constructive feedback.

We have taken the time to address all queries and prepared a response letter and a revised manuscript. Author **replies** to point-by-point comment and **changes** made to the manuscript are highlighted in **red** and **blue**.

To Reviewer 1:

In this manuscript, the authors applied nitrate salts/H₂O system as a model to study the ionic conductivity variation by concentration and temperature. Based on the data from previous reports, the authors proposed a new model to emphasize this relationship, where the accuracy is over 95%. However, the authors did not clarify the relationship between ionic conductivity variation and solvation structure difference. Furthermore, previous researchers have studied similar models, but the authors did not differentiate their model from others', e.g., accuracy, benefits, etc. Therefore, the paper can not be considered for publishing in CommsChem until the author appropriately addresses the following issues:

Point 1: The authors selected three different types of eutectic systems to study the ionic conductivity of nitrate solutions. However, all the K_{max} is at the eutectic composition and the conductivity decreases afterward. Therefore, will the number of eutectic points and solvation structures affect this trend?

Author reply: Thank you for your comment. As written in the manuscript on line 248. We only witness a common confirmed correlation for the first eutectic point only. We found this to be a common feature for most liquid electrolytes with some exceptions such as sulphuric acid that shows a second K_{mx} at the third eutectic point. This is currently under investigation by our research team to evaluate whether the phase diagram is actually a mirror image of conductivity isotherms along with the role of eutectic points/solvate forming points on conductivity.

Change made to the manuscript: We have added a line to clarify this. "For most of the studied electrolyte solutions, it is found that subsequent eutectic points and solvate formation points do not have a unique common effect on the room temperature conductivity isotherm"

Point 2: Does all the experimental data (except LiNO₃) come from previous reports? Did the authors get the original data and how accurate are they?

Author reply: As indicated in the experimental section starting from line 130. We have obtained conductivity data from the mentioned sources. The form of the raw data is also written and how they were converted to mS cm^{-1} is clearly written in step by step. Most data were either obtained as raw numerical data or digitized from figures. We have included the name of digitization software we used on line 132 (Automeris software). Data are 100% accurate.

Change made to the manuscript: None.

Point 3: At eutectic points, all the systems demonstrate the highest conductivity and the trend is H to Ca. As the authors claimed, it is due to the charge density of cations. However, protons use a Grotthuss mechanism to transport instead of a Viechle mechanism. Therefore, are they compatible?

Author reply: You are right and we have included this in the discussion already on line 374 where we justified the very high conductivity for nitric acid relative to the other solutes.

Change made to the manuscript: None.

Point 4: The author assumed that the anion would not affect the conductivity. Is there any evidence? Moreover, the concentration of the cation varies a lot in different systems. Will the cation concentration affect the conductivity? At the same concentration (dilute), will the ionic conductivities follow this trend?

Author reply: We have not made any assumption in regards to the anion role. It is well established in literature that the anion does not affect conductivity due to its large size and low energy of interaction with solvents (see <https://doi.org/10.1039/c2fc9ra07824j>). We have also included NO_3^- charge density and interaction energy on line 356 as an example. In regards to the cation concentration, we would like to thank you for bringing this point up to our attention, we will use it to enrich the discussion. To answer your question, we still believe that the charge density and its impact on ion concentration and their mobility has the dominant effect on κ_{max} . However, cation concentration could be the reason that calcium nitrates x_{max} is almost half of lithium nitrate's x_{max} despite the close charge density values.

Change made to the manuscript: added a footnote to table 1 that reads: “**only divalent solute; provides double cation concentration at any given concentration”

Point 5: For the modeling part, most of the content and equations are similar to previous papers, such as ACS Omega 2020, 5, 22465–22474. The authors applied the Arrhenius equation to bring in the temperature factor. What is the benefit of this model? Are they different from previous reports? It seems like combining several constants to A and B. The authors should provide more information.

Author reply: ACS Omega 2020, 5, 22465–22474 reference have arrived to their $\kappa = f(x, T)$ equation upon building on equations published by our group before (see DOI 10.1149/2.1601704jes) which explain the similarity in the derivation of their equation and ours

which ultimately serve the same purpose of introducing T. This is why we included a model comparison at the end and explained that the value of our equation lies in the reduced number of parameters. The importance of fewer parameters is mentioned in line 570-579 in the discussion

Change made to the manuscript: None

Point 6: Does this equation fit all systems? Unfortunately, the Ca one does not fit very well. Also, except nitrate system, will changing anions influence this model?

Author reply: It is true that this equation does not fit calcium nitrate the best. Yet, it is still acceptable fitting as determined by the coefficient of determination R^2 . We are currently investigating different electrolyte solutions based on their chemistry (e.g. acids, bases, ...) to verify the applicability of the equation to each chemical family- a huge undertaking indeed.

Change made to the manuscript: None.

Point 7: The authors should enrich the description for the figure captions. They are not very clear. Additionally, from Figure 3 to Figure 7, they are mismatched with the content in the main text. Please correct them.

Author reply: Thank you for pointing this out. Change was made as requested.

Change made to the manuscript: We have revised each of the figures' captions and changed the unclear ones. We also fixed any mismatches that occurred in text

Point 8: Please carefully check the paper. Multiple typos, grammar, and reference issues.

Author reply: Thanks for the comment. We performed a language test along with detailed double checking of all data, figures and references. We found several subtle none fundamental mistakes and fixed them all.

Change made to the manuscript: We have done an overall spelling, grammar, data and references check. Ref 26 is updated. Figures 3,4,5,6, and 7 were updated.

To Reviewer 2:

In the present work, Al-Salih and co-workers investigated the transport behavior of aqueous solution containing different cations, and attempted to connect the structural changes in the phase diagrams with ionic conductivity. The results are quite interesting and the manuscript is well organized, I would therefore recommend the publication of this work after addressing the following minor issues.

Point 1: Figure 1. To improve the readability of this figure, it is recommended to further illustrate the concepts of vehicular and ion-hopping mechanisms.

Author reply: Thank you for the recommendation.

Change made to the manuscript: We have added the following lines before the figure to clarify these concepts. "...diffusing or migrating by a vehicular mechanism through free volume to a loose lattice structure where transport is dominated by ion hopping mechanism where naked ions hop along the extended structure (different types of ICs) from one site to another energetically favorable vacant site (free volume) (Fig 1). Both mechanisms often depend on temperature, as higher temperatures can provide the energy needed for ions to overcome energy barriers and move through the free volume that in turn can expand with temperature."

Point 2: Line 142. The figure presented does not follow the numeric order. Please revise it.

Author reply: Thank you for pointing this out.

Change made to the manuscript: We have fixed the numerical order

Point 3: Lines 148-152. It is not clear why the conductivity values reported by Li et al. differ from those reported in this work.

Author reply: We had two reasons to doubt the results reported by Li et al. The first, is the presence of two local maxima points in the conductivity isotherms which is inconsistent with the paper findings. The second is the discrepancy of the data with other sources in literature (see: <https://doi.org/10.1139/v55-184>). We had to do our own experiments to come up with reliable data. This was not the case for the other nitrates where we had multiple sources showing identical data and we just had to choose the source with most data points.

Change made to the manuscript: None

Point 4: The transport of ionic species is tightly associated with the ligand exchange rate (i.e., Eigen value), as discussed in recent work (Energy & Environmental Science 2023, 16, 11). The authors are recommended to include this point in the revised version.

Author reply: Thank you. We have now added this point in the introduction.

Change made to the manuscript: added the following line: "The energy barriers are directly linked to the charge density of the ions and their ability to attract and retain solvent molecules that can be quantified by the Eigen number in the case of water as a solvent. [9,10]"

To Reviewer 3:

Al-Salih et al. have studied the ionic conductivity (κ) as a function of molar ratio (x) and Temperature (T) for aqueous solutions of salts with nitrate anion and different cations (proton, lithium, calcium, and ammonium) along with their liquid-solid phase diagrams by Semi-Empirical Modeling. The idea is interesting, but some problems are still confusing. Questions are shown as following:

Point 1: The authors propose four systems such as Ammonium nitrate/water and Lithium nitrate/water etc. The authors think that Ammonium nitrate/water system has a simple eutectic and

no solvate formation and a conductivity isotherm with a maximum conductivity that drops slightly thereafter etc. However, I wonder how the authors ensure these systems have their proposal properties.

Author reply: We have thoroughly searched the literature for phase diagrams and conductivity isotherms of the electrolyte solutions and selected the ones that were consistent from different sources. The selected phase diagrams/conductivity isotherms were directly digitized. This is mentioned at the end of the experimental section on line 158.

Change made to the manuscript: None

Point 2: The authors use their own tested data for LiNO₃, but other data are obtained from the references, such as NH₄NO₃, HNO₃, Ca (NO₃)₂. I really worry that these data can not support their proposal ideas. The authors should better use their own data.

Author reply: Data for NH₄NO₃, HNO₃, Ca (NO₃)₂ were obtained from literature where we found multiple sources reproducing the same data for each. However, for LiNO₃, we had reasons to doubt the results reported by Li et al. see our reply to point 3 by reviewer #2 We had to do our own experiments to come up with reliable data. This was not the case for the other nitrates where we had multiple sources showing comparable data and we just had to choose the source with most data points to improve the quality of the model.

Change made to the manuscript: None

Point 3: Some illustrations don't match the Figures. For example, Fig 3a-d should be Figure 2a-d in page 6. The authors should check the details throughout their manuscript.

Author reply: We have checked the figure numbering and in-text references to figures and made sure they are now correct.

Change made to the manuscript: Updated figure numbers in captions and in-text.

Point 4: I confuse how the authors obtain the phase diagram in Figure 2, through calculation or experiments? The specific details should be provided.

Author reply: They are experimental data reported by others as we responded in point 1

Change made to the manuscript: None

Point 5: The authors think that their formular can apply to high concentration solutions, however, the molar ratio is less than 0.5 as shown in Figure 5 and 6. It should be noticed that the realistic concentration of electrolyte is very high (> 1 mol/L) in batteries, so I wonder how the authors think about higher concentration of electrolyte.

Author reply: We have actually fixed the term as we actually have calculated molar fraction of solute in solvent and this is what we meant by x throughout the manuscript. We have changed this in all of the text.

Change made to the manuscript: change “molar ratio” to “molar fraction” throughout the manuscript

We have also concluded our own thorough review and have slightly edited the manuscript accordingly. Thank you and looking forward for your positive reply.

Sincerely yours,

Dr. Yaser Abu-Lebdeh
Senior Research Officer
Battery Materials Innovation Team Lead
Energy, Mining and Environment Research Centre
National Research Council Of Canada
1200 Montreal Road, Ottawa, Ontario K1A 0R6
Tel: 1 613 949 4184; Fax: 1 613 991 2384
E-mail: Yaser.Abu-Lebdeh@cnrc-nrc.gc.ca

REVIEWERS' COMMENTS:

Reviewer #1 (Remarks to the Author):

The authors have addressed all my concerns. Therefore, I recommend this paper to be published.

Reviewer #2 (Remarks to the Author):

My previous concerns have been clarified in the revised version, I would therefore recommend the publication of this work.

Reviewer #3 (Remarks to the Author):

The revised manuscript can be considered for publication if other reviewers' comments have also been well addressed.